# Characterization of Acidic Mammalian Chitinase as a Novel Biomarker for Severe Periodontitis (Stage III/IV): A Pilot Study

**DOI:** 10.3390/ijerph19074113

**Published:** 2022-03-30

**Authors:** Ming Yang, Yunjo Soh, Seok-Mo Heo

**Affiliations:** 1Department of Periodontology, School of Dentistry, Jeonbuk National University, Jeonju 54907, Korea; yangming7279@sina.com; 2Department of Periodontology, School of Dentistry, Beihua University, Jilin 132013, China; 3Laboratory of Pharmacology, Institute of New Drug Development, School of Pharmacy, Jeonbuk National University, Jeonju 54907, Korea; ysoh@jbnu.ac.kr; 4Research Institute of Clinical Medicine of Jeonbuk National University, Jeonju 54907, Korea

**Keywords:** AMCase, enzyme-linked immunosorbent assay, periodontitis, biomarker

## Abstract

Periodontitis is a chronic inflammatory condition characterized by gingival infection, periodontal pocket formation, and alveolar bone loss. Acidic mammalian chitinase (AMCase), an active chitinase enzyme, increased its expression under severe inflammation and related systemic disorders. However, AMCase expression and molecular mechanism in periodontal inflammation, have not been elucidated yet. This study was aimed to characterize AMCase in severe periodontitis patients compare to those in periodontally healthy subjects. In total, 15 periodontally healthy subjects and 15 severe (stage III/IV) periodontitis patients were enrolled with their informed consent. Tissue samples were collected and analyzed using Western blot and enzyme-linked immunosorbent assay (ELISA). AMCase protein expressions in periodontal patients were significantly more increased than those of periodontally healthy individuals. ELISA resulted in median values (first quartile to third quartile) of the periodontally healthy group 0.654 ng/mL (range, 0.644–0.827 ng/mL) and the periodontitis group 0.965 ng/mL (range, 0.886–1.165 ng/mL). AMCase was expressed significantly higher levels in periodontitis patients than in periodontally healthy individuals (*p* < 0.05). This suggests that AMCase may play a potential role as a biomarker for the screening and early diagnosis of severe periodontitis.

## 1. Introduction

Periodontal disease mainly involves bacterial infection and host inflammatory processes, resulting in periodontal tissue damage, alveolar bone loss, and eventual loss of teeth [1,2]. Two major types of periodontal disease are periodontitis and gingivitis [1,3]. Accordingly, periodontal disease is considered one of the most common diseases globally, affecting up to 95% of the population [1]. In particular, the prevalence of severe periodontitis ranges from 5% to 20% [1]. Periodontitis is caused by an imbalance between symbiotic normal flora and dysbiotic biofilms [2,4]. Moreover, complex pathogenicity is mainly affected by host immunological dysregulation, depending on various genetic, habitual, and environmental risk factors [2].

The complexity of periodontal disorders has confronted researchers and clinicians with considerable challenges in the correct diagnosis and classification of periodontal diseases. Classical diagnosis of periodontitis has been conducted based on clinical parameters, such as periodontal pocket probing, bleeding on probing (BOP), plaque index, and clinical attachment levels (CALs) [3,5]. In recent years, however, a new periodontal classification was developed by multi-dimensional staging and grading systems related to biologic and pathologic influences on both oral and systemic diseases [6,7].

Numerous studies have suggested that oral diseases are closely associated with systemic diseases, including cardiovascular disease [8], diabetes mellitus [9], and respiratory disease [10]. In order to estimate the potential effect of periodontitis on systemic disease, various potential biomarkers have been investigated using body fluid, such as saliva, serum, gingival crevicular fluid, and periodontal pocket tissue [11,12,13]. Oral pathogens in periodontal pocket tissue may access other systemic organs through blood vessels [8,9] or respiratory tract [10]. In addition, inflammatory mediators of periodontitis can exacerbate systemic disease by activating inflammatory biomarkers, such as IL-1, TNF-α, and C-reactive proteins [9,14,15]. 

Chitinases are hydrolytic enzymes that dissipate glycosidic bonds in chitin. As the next most copious polysaccharide in environment, chitin is known to provide structure and rigidity to fungal cell walls [16,17]. Although humans and mice are unable to synthesize chitin, chitinases can be produced [16]. Indeed, two active chitinases, chitotriosidase (Chit1), and acidic mammalian chitinase (AMCase), are found in humans and mice [17,18,19]. AMCase has attracted immense attention for a novel biomarker due to increasing findings demonstrating its expression under specific pathological conditions and chronic severe inflammations, including gastric cancer, Crohn’s disease, and a chronic obstructive pulmonary disease [18,20,21].

In the study of periodontal diseases, chitinases have also generated considerable attention since human chitinase can play an important role in the defense against chitin-containing oral pathogens [22,23]. However, the exact function or related mechanism of AMCase in periodontitis has not been investigated. So far, chitinase activity in periodontitis has been measured solely in saliva [22,23] and the existence of AMCase protein from periodontal pocket tissue, has not been identified in human. Since periodontal pocket is the anatomo-pathological entry lesion to initiate periodontal inflammation [13], AMCase in periodontal tissue may provide a substantial information as a site-specific biomarker for periodontitis.

Thus, this study was aimed to characterize AMCase in severe periodontitis patients compare to those in periodontally healthy subjects. As a null hypothesis, periodontal inflammation does not affect the level of AMCase. 

## 2. Materials and Methods

### 2.1. Study Participants

Participants were recruited for this clinical study under Institutional Review Board approval (No. 201712023, date of approval: 19 April 2018) of Jeonbuk National University Dental Hospital (May 2018–April 2019). The completed Strengthening the Reporting of Observational Studies in Epidemiology (STROBE) checklist is shown in Appendix A. Subjects between 30 and 70 years of age enrolled in our study voluntarily and provided written informed consent after comprehensive dental examinations. All participants did not take any anti-inflammatory agents and antibiotics at least 4 weeks before enrollment. The exclusion criteria were as follows: systemic disorders affecting their periodontal conditions, including cardiovascular disease, respiratory diseases or diabetes mellitus (HbA1c ≥ 7%). Smokers were not eliminated, but their smoking history was recorded. A total of 30 subjects were selected for this study (Figure 1a). Among them, 15 patients (*n* = 15) were diagnosed with severe periodontitis: stage III (*n* = 10)/stage IV (*n* = 5) according to the 2017 periodontal classification [6,7] (Table 1). All 15 periodontitis subjects received scaling and root planing before periodontal flap surgery. From each patient receiving periodontal flap surgery, a periodontal pocket tissue indicating deep probing pocket depth (>=6 mm) was eliminated and collected for sampling (Figure 1b, right panel). From all the 15 periodontally healthy subjects who received implant surgery, each sample tissue was collected from a edentulous site for implant surgery by a tissue punch drill (Ø = 4.8 mm, Osstem, Seoul, South Korea) (Figure 1b). All tissue samples were collected by a periodontal specialist in the Department of Periodontology at Jeonbuk National University Dental Hospital. 

### 2.2. Western Blot Analysis

Immediately after sample collection, periodontal tissues were washed with phosphate-buffered saline (PBS) five times and cut into pieces with a sterilized scissor. Tissue pieces were washed with Gey’s balanced salt solution (Sigma) for 30 min at room temperature to remove erythrocyte. Then, tissue pieces were lysed in an ice-cold lysis buffer (1 mM phenylmethylsulfonyl fluoride (PMSF), 0.25% sodium deoxycholate, 1 mM Na3VO4, 1% NP-40, 1 μg/mL pepstatin, 1 μg/mL leupeptin, 1 mM EDTA, 5 μg/mL aprotinin, 20 mM NaF, 150 mM NaCl, 50 mM Tris-HCl), and kept on ice for 1 h. Tissue pieces were centrifuged at 13,200 rpm at 4 °C for 10 min. Protein separation was performed by 10% SDS-PAGE. Membrane was transferred in Polyvinylidene difluoride membrane and blocked (5% non-fat skim milk) in Tris-buffered saline with 0.25% Tween 20 (TBST) at 16 °C for 1 h. The diluted anti-AMCase (Santa Cruz, CA, USA) was incubated for 18 h at 4 °C and was diluted as 1:2000–1:3000 in 5% skim milk in TBST. The membrane was incubated by secondary antibodies in 5% skim milk (1:2500–1:3000) for 2 h. Blots were then developed by Enhanced Chemiluminescence blot detecting agent (Amersham Biosciences, Piscataway, NJ, USA).

### 2.3. Enzyme-Linked Immunosorbent Assay (ELISA)

The human CHIA (Acidic mammalian chitinase) ELISA kit (MyBioSource, San Diego, CA, USA) was used to determine the levels of AMCase. All tissue samples from periodontal pocket tissue and edentulous sites were lysed in an ice-cold lysis buffer (150 mM NaCl, 50 mM Tris-HCl, 0.25% sodium deoxycholate, 1% NP40, 1 μg/mL aprotinin, 1 mM phenylmethylsulfonyl fluoride (PMSF), 1 mM EDTA, 1 μg/mL pepstatin, 1 μg/mL leupeptin). The suspension was sonicated with an ultrasonic cell disrupter to break the cells further. Tissue pieces were centrifuged at 13,200× *g* at 4 °C for 10 min, and the supernatants were collected for the ELISA assay following the manufacturer’s instructions.

### 2.4. Statistical Analysis and Sample Size Determination

Statistical analyses were performed by Student’s *t*-test and one-way ANOVA with Tukey’s multiple comparison by SPSS 12.0. All experimental data are presented as mean ± S.E.M. Statistical significance was indicated if *p*-values less than 0.05. The effective sample size was 30 (15 in each case) under 0.8 power calculation (effect size 0.40, α = 0.05) by the G-Power 3.1 program (Dusseldorf, Germany). 

## 3. Results

### 3.1. Characteristics of Periodontally Healthy Subjects and Periodontal Patients

Of the 30 subjects who voluntary participated in this clinical study did not have systemic diseases, including cardiovascular disease, respiratory disease, acquired immunodeficiency syndrome, pregnancy, and breastfeeding. After comprehensive periodontal examination, a total of 30 subjects were selected and divided into two groups under the 2017 World Workshop Periodontal Classification [7]: a periodontally healthy group and a severe periodontitis (stage III or IV) group (*n* = 15 each) (Figure 1). Both groups were slit into categories of gender, smoking, age, clinical attachment loss, bleeding on probing pocket depth, as shown in Table 1. 

### 3.2. Higher Level of AMCase in Severe Periodontal Patients

In order to observe the expression of AMCase in periodontally healthy individuals and severe periodontitis (stage III or IV) patients, human tissue samples were collected and analyzed by Western blot. Samples from periodontally healthy subjects were collected from the edentulous tissue site (Figure 1). Samples of severe periodontitis patients were taken from periodontal pocket tissue when probing pocket depth (PPD) is more than 6 mm (Figure 1a). In all cases, human AMCase protein was identified in periodontal tissue at 50 kD after using a specific anti-AMCase antibody and probing with Western blot (Figure 2a). However, the levels of AMCase in the samples from the periodontitis patients were significantly (*p* < 0.05) higher than those of periodontally healthy subjects (Figure 2b). These results suggest that AMCase of the periodontal inflammation is higher than that of periodontally healthy condition. 

### 3.3. Comparison of AMCase Levels between the Periodontally Healthy and Periodontitis Groups

In order to investigate the relationship between healthy edentulous site tissue and periodontitis pocket tissue, the concentrations of AMCase for both groups were measured by ELISA. The results demonstrated AMCase concentrations from periodontal tissue in both the periodontally healthy and periodontitis group (Figure 3). Table 2 shows the median values for the periodontally healthy group 0.654 ng/mL and the periodontitis group 0.965 ng/mL. Moreover, it indicates the 25th to 75th percentiles for the periodontally healthy group range 0.644–0.827 ng/mL and the periodontitis group range 0.886–1.165 ng/mL. The difference in AMCase levels between the periodontally healthy and periodontitis groups were statistically significant (*p* < 0.05) by Student’s *t*-test. These results suggest that AMCase concentrations in the periodontal tissue were higher in periodontal patients in comparison with periodontally healthy subjects.

## 4. Discussion

In this clinical study, we divided all participants into two categories: periodontally healthy group and severe periodontitis group (stage III/IV) under the 2017 World Workshop Periodontal Classification [7] (Figure 1, Table 1). This recent classification is aimed at characterizing the biological features of periodontitis and related systemic diseases by addressing the initial framework to introduce potential biomarkers [6,7]. Many studies have found that numerous pro-inflammatory biomarkers in periodontitis also increase with chronic inflammatory diseases [9,14,15,24]. However, reliable biomarkers or functional profiles for periodontal classification have not been adequately defined. In the present study, we demonstrated that AMCase protein expression was significantly increased in the severe periodontitis group (stage III/IV). To the best of our knowledge, this is the first study to assess human AMcase as a novel and original marker for periodontal disease.

AMCase, an active enzyme of chitinase, belongs to a family of 18 glycosyl hydrolases found in humans and mice [17,18,19]. Flach et al. [25] proposed that chitinase plays a role in the hydrolysis of chitin-containing pathogens, such as bacteria and fungi, because these chitin coats protect pathogens from the harsh environment inside the host [26]. Paradoxically, while chitin does not exist in human tissues, the chitinase family, including AMCase, has been detected in humans [17]. Human AMCase has recently attracted attention because AMCase substantially increases tissue expression under chronic inflammation in both the lungs and stomach [16,20]. For example, AMCase levels increase significantly in ocular allergy, dry eye syndrome, and gastric cancer [20,27,28]. Other studies have also shown that AMCase mRNA and protein levels are significantly increased in chronic inflammatory lung diseases, such as asthma and chronic obstructive lung disease [16,18].

In recent dental investigations, some studies have been performed to detect chitinase activity in the saliva [22,29]. However, whether chronic inflammation of periodontal tissues directly or indirectly affects AMCase levels remains to be determined. In the present study, we found that AMCase expression was increased in periodontal inflammatory tissues. AMCase protein levels in the tissues of periodontally healthy individuals and patients with periodontitis were measured using specific antibodies to identify AMCase (Figure 2). AMCase is one of the major active chitinases containing 12 exons, which transcribe a 50 kD protein [17,27]. In our study, human AMCase protein from periodontal tissues was also characterized at 50 kD in accordance with previous studies [17,27]. In other investigations, AMCase mRNA and protein levels have been characterized in the epithelial cells of lung tissue, as well as in the gastric mucosa during chronic inflammation [18,28]. Accordingly, similar expression profiles of AMCase from different human organs suggest that AMCase may be highly tissue-specific in periodontal tissue, with common functions remaining unknown. Recently, anti-AMCase drugs have been investigated for the treatment of chronic inflammation [30,31]. Since new potential drugs have also targeted the periodontal area [32], novel therapeutic drugs, such as AMCase inhibitors, would shed light on the next strategy of periodontal research.

In the present study, AMCase protein expression in patients with periodontal disease was compared to that in periodontally healthy individuals. Therefore, a shortcoming of this study was that a direct comparison between the same harvest sites could not be implemented. Due to ethical concerns under the IRB in this study, collection of healthy gingiva or periodontal tissue from healthy subjects was prohibited. In the future, it would be helpful to confirm these results by obtaining healthy tissue samples from adjacent infected areas in the same individual or at the same location in the animal periodontitis model. Alternatively, non-invasive sampling approaches have been suggested. For example, various samples from the oral cavity can be harvested from different anatomical sites, including the saliva, dental plaque, and gingival crevicular fluid. 

Another limitation of this study was the evaluation of AMCase levels in patients with severe periodontitis alone (III/IV). Thus, this study did not include patients with gingivitis or mild periodontitis in the intermediate stages between healthy and severe periodontitis. In our pilot study design, the sample size was too small to include all stages or grades of periodontal disease. In the future, it will be necessary to investigate the role of AMCase and its biological mechanisms across all the stages of periodontitis. Moreover, randomized controlled or intervention studies in larger cohorts would be meaningful in the future.

## 5. Conclusions

Our study investigated a new therapeutic target for periodontal inflammation since human AMCase increased its expression under severe inflammation related systemic disorders. This human study characterized AMCase protein from periodontal pocket tissue from periodontitis. Our study also demonstrated that the concentration of AMCase was significantly increased in the severe periodontitis group. Therefore, AMCase may act as one of novel potential biomarkers for screening and diagnosis of severe periodontitis (stage III/IV). 

## Figures and Tables

**Figure 1 ijerph-19-04113-f001:**
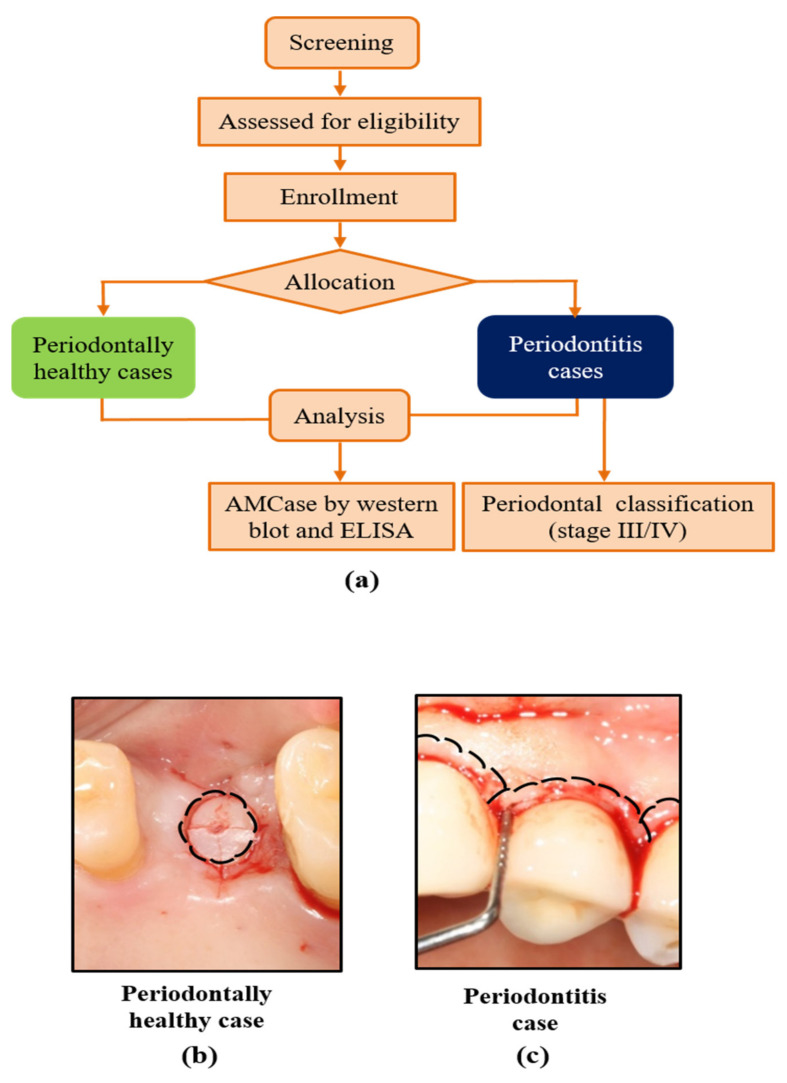
Flow diagrams of this study and example cases for sample collection. Each clinic visit consisted of a basic oral examination including probing depth, clinical attachment and bleeding on probing. All subjects between 30 and 70 years participated voluntarily this clinical study, under informed consent. For periodontitis cases, stages and grades were divided by the 2017 World Workshop Periodontal Classification (**a**). Periodontally healthy tissue samples (dotted lines of circle) were collected from edentulous area by tissue punch during implant surgery (**b**). Periodontitis tissue samples (dotted lines of arc) were collected from periodontal pocket area during periodontal flap surgery (**c**). AMCase: acidic mammalian chitinase, ELISA: Enzyme-linked immunosorbent assay.

**Figure 2 ijerph-19-04113-f002:**
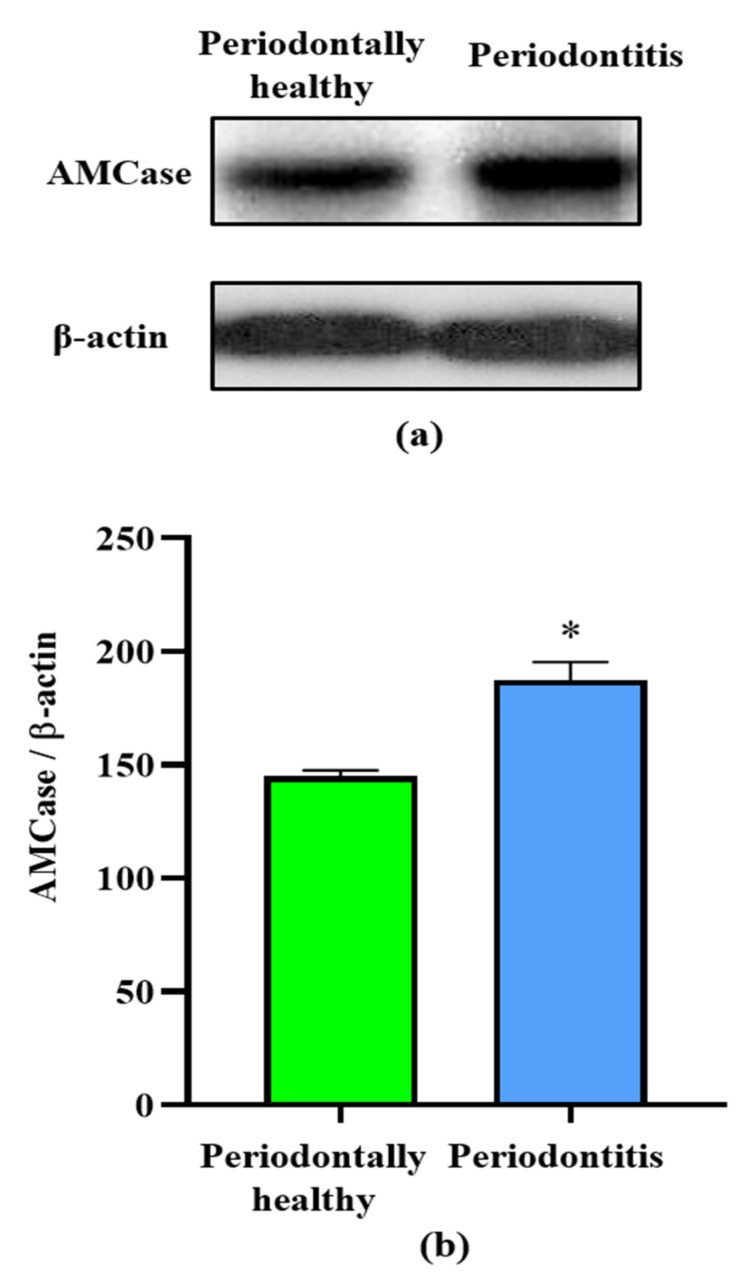
Analysis by Western blot of AMCase in human tissues from representatives of periodontally healthy and periodontitis cases. Tissues were lysed in lysis buffer, protein levels of AMCase were expressed by Western blot with specific antibodies against AMCase and β-actin (**a**). The histogram demonstrated AMCase protein expression level (%) compared to control (**b**). Severe periodontitis groups were compared with periodontally healthy groups (* *p* < 0.05). The results were described as mean ± S.E.M. Significant differences were determined by one-way ANOVA with Tukey’s multiple comparison.

**Figure 3 ijerph-19-04113-f003:**
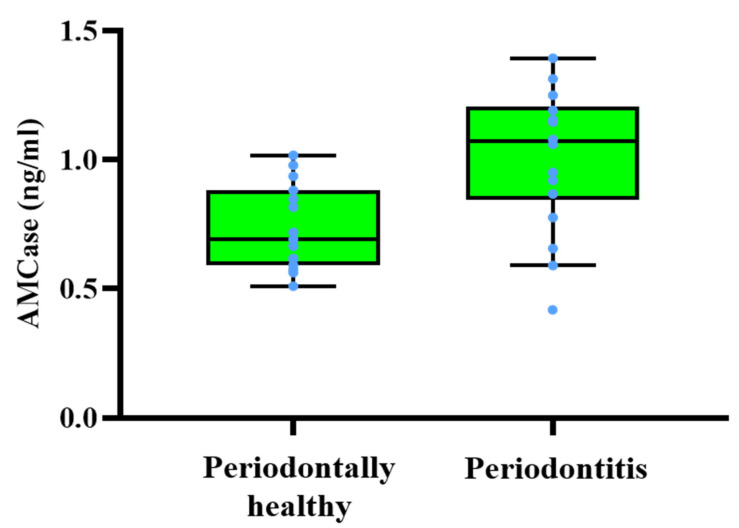
AMCase levels of human tissue from periodontally healthy and periodontitis groups. AMCase human tissue concentrations were determined by ELISA (enzyme-linked immunosorbent assay system). Periodontal patients were demonstrated a greater amount of AMCase in periodontal tissues than those in the tissues from periodontally healthy group. The median values (Q1–Q3) from periodontally healthy case and periodontitis case were 0.654 ng/mL (0.644–0.827 ng/mL, range) and 0.965 ng/mL (0.886–1.165 ng/mL, range). The difference in AMCase levels between the periodontally healthy and periodontitis groups was statistically significant (*p* < 0.05). The central box indicates from Q1 to Q3. The middle line in each box indicates median. Outlier values were set if they are larger than the 90th percentile or small than the 10.

**Table 1 ijerph-19-04113-t001:** Characteristics of the clinical study groups.

Characteristics	Periodontally Healthy Group	Periodontitis Group	*p*-Value
Total subjects, N (%)	15 (50)	15 (50)	
Gender, N (%)			0.011
Male	13 (86.7)	6 (40)	
Female	2 (13.3)	9 (60)	
Smoking, N (%)			0.500
Smokers	2 (13.3)	1 (6.7)	
Nonsmokers	13 (86.7)	14 (93.3)	
Age, M ± SD (yr)	62.00 ± 7.59	58.13 ± 10.72	
PPD (mm)	N/A (edentulous)	4.18 ± 0.72	<0.001
CAL (mm)	N/A (edentulous)	1.32 ± 0.64	<0.001
BOP (%)	N/A (edentulous)	66.93	<0.001

Values are presented as number of patients (%) or mean ± standard deviation (M ± SD). N/A: not applicable, PPD: probing pocket depth, CAL: clinical attachment level, BOP: bleeding on probing.

**Table 2 ijerph-19-04113-t002:** AMCase levels in the periodontally healthy and periodontitis groups.

Characteristics	Periodontally Healthy Group	Periodontitis Group	*p*-Value
Total subjects, N (%)	15 (50)	15 (50)	
Mean ± SD (ng/mL)	0.736 ± 0.165	1.026 ± 0.24	< 0.05
Standard error of the mean	0.043	0.065	
Median (ng/mL)	0.654	0.965	
25th to 75th percentiles (ng/mL)	0.644–0.827	0.886–1.165	

AMCase: acidic mammalian chitinases. Values are presented as number of patients (%) or mean ± standard deviation (M ± SD).

## Data Availability

Supporting data and results can be found in the tables and figures.

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
