# Peer review of "Characterization of Acidic Mammalian Chitinase as a Novel Biomarker for Severe Periodontitis (Stage III/IV): A Pilot Study"

_ijerph, 2022, doi:10.3390/ijerph19074113_

Round 1
Reviewer 1 Report
Dear Authors, I read your article submitted in MDPI-IJERPH. Below are my comments.
GENERAL CONCERNS
- Please check English grammar. Look for misprints.
TITLE AND ABSTRACT
- The title of the manuscript conveys with the major concern of the study.
- The abstract properly summarize the topic addressed.
INTRODUCTION
- The null hypothesis is missing. Please set it at the end of the Introduction section, following the aim of the study.
MATERIALS AND METHODS
- Why did you choose to not include smoking into exclusion criteria?
- To me, the harvest method is a weak point of the study. It does not ensure fair comparison: it is not possible to compare different harvest site with different histological feature between the two groups. The harvest site should be the same. I know the struggle for the ethical concern, but it cannot be a justification. It exposes to scientific unsoundness. Moreover, it should be more appropriate to include, as it is the first study to evaluate the molecular function of AMCase in periodontal tissues, also patients with gingivitis to investigate the role of the protein in different conditions. Please clarify.
- Pag.4 line 144: “as shown in…” Table 1 is missing.
- In the statistical analysis section you need to include the Student t test, too.
DISCUSSION
- Pag 7 lines 207-209: “The aim of our study was to compare the levels of AMCase in periodontal patients’ periodontal pocket tissue with those in periodontally healthy subjects’ edentulous tissue.” It is a repetition, please remove it.
- In general, Discussion section should analyse the findings of the study in a broader context, comparing to other study results, or give speculation. If your study as you stated id the first on the theme, the Discussion section should be more incisive and detailed. Give speculations. Cite article for example about the pathogenetic path in which the protein could be involved, and so on. In the present manner, Discussion section is not appropriate at all. Also, study’s limitations and future research are missing.
CONCLUSIONS
- Do not include citation in Conclusion section. Move lines 225-228 Pag. 7 to the Discussion section. Synthetize relevant conclusion basing on your findings.
Reviewer 2 Report
Title:Characterization of acidic mammalian chitinase as a novel biomarker for severe periodontitis (stage III/IV)
Manuscript ID: ijerph-1636707
The manuscript is novel insofar as an original marker of inflammation in periodontitis is investigated. However, it needs to be improved in different aspects that are described below. The methodological part of the manuscript leaves many important issues and gaps unanswered.
Introduction.
-The authors state: "with an incidence of 31 5% to 20% ". Incidence is a rate, therefore it is expressed per unit of time. I believe that the authors want to give a prevalence of severe forms of periodontitis, because these data do not correspond to the prevalence of periodontitis in the general population. Clarify.
-The authors recognize the complexity of periodontal disorders, but provide a very simplistic definition of periodontitis. They do not include the role of a dysbiotic biofilm, immunological dysregulation, and the other environmental risk factors... Include a more complete definition.
-Penultimate paragraph, line 65: "....and other systemic diseases." What other systemic diseases can benefit from a certain marker in the periodontal pocket? Clarify.
-Last paragraph, lines 68-70. This is a conclusion and not of this section.
Material and Methods.
-In an observational study, a STROBE check list, it is desirable to perform.
-Being a study on an inflammatory marker, how the patients who took anti-inflammatories and/or antibiotics in recent weeks not been excluded?.
-The authors say they have followed the periodontal classification of 2017. In it, clearly specified how the tobacco consumption, modifying factor, should be collected. As the authors collect it, this variable is undervalued.
-In all investigations where the variables are measurements, the exploring researcher must be calibrated to demonstrate to readers that they were done correctly. Kappa or concordance indices are necessary.
-Moderate periodontitis, does not correspond to a stage III, IV. They are severe and advanced forms of periodontitis. Correct.
-With what type of periodontal probe was the examination performed?.
-Statistical analysis and sample size determination. How do the authors claim to assume a p ≤ 0.05 in the statistical tests, while for the calculation of the sample size they assume an alpha or type 1 error so undemanding, 0.2?.
-any reason to use such an old version of SPSS?.
Results
-The authors define their study as "controlled clinical study". Did they perform an adjusted multivariate analysis to control for confounding variables? They must correctly define the epidemiological design of the study.
-The tables must carry at the bottom of the table, the statistical test performed. In the text, the authors say that they compared the ELISA results between the two groups, with T-student, were the normality of the variables checked to apply this test? The sample of authors is very small.
Discussion
-The first two paragraphs of the discussion should be summarized or removed. They repeat the objective of the study, and its content seems more like an introduction.
-The third paragraph, looks like a results content.
-While the conclusion must be the succinct response to its objectives. It is not the place to make comparisons.
- In the opinion of this reviewer, the discussion should be redone. If there are no studies to compare with, the authors should interpret their results in more detail, the implications and applicability of their study, the limitations..........
-Given the small sample size of the study, would the authors be willing to call it a pilot study? .
Round 2
Reviewer 1 Report
No any other comments.
Author Response
Thank you so much. We appreciate it.
Reviewer 2 Report
Just one last clarification. In the introduction, the authors describe a prevalence of periodontal diseases of up to 95%. This reviewer understands that the authors refer to the sum of gingivitis and periodontitis. If so, the authors should word this paragraph more clearly.
Author Response
Thank you for your advice. Accordingly, we changed it in the first paragraph of the Introduction as you recommended.